# Novel Immunotherapies and Combinations: The Future Landscape of Multiple Myeloma Treatment

**DOI:** 10.3390/ph16111628

**Published:** 2023-11-19

**Authors:** Sonia Morè, Laura Corvatta, Valentina Maria Manieri, Erika Morsia, Antonella Poloni, Massimo Offidani

**Affiliations:** 1Clinica di Ematologia Azienda Ospedaliero Universitaria delle Marche, 60126 Ancona, Italy; 2U.O.C. Medicina, Ospedale Profili, 60044 Fabriano, Italy

**Keywords:** relapsed multiple myeloma, bispecific antibodies, teclistamab, elranatamab, talquetamab, cevostamab

## Abstract

In multiple myeloma impressive outcomes have improved with the introduction of new therapeutic approaches, mainly those including naked monoclonal antibodies such as daratumumab and isatuximab. However, moving to earlier lines of therapy with effective anti-myeloma drugs led to an increase in the number of patients who developed multi-refractoriness to them early on. Currently, triple- or multi-refractory MM represents an unmet medical need, and their management remains a complicated challenge. The recent approval of new immunotherapeutic approaches such as conjugated monoclonal antibodies, bispecific antibodies, and CAR T cells could be a turning point for these heavily pretreated patients. Nevertheless, several issues regarding their use are unsolved, such as how to select patients for each strategy or how to sequence these therapies within the MM therapeutic landscape. Here we provide an overview of the most recent data about approved conjugated monoclonal antibody belantamab, mafodotin, bispecific antibody teclistamab, and other promising compounds under development, mainly focusing on the ongoing clinical trials with monoclonal antibody combination approaches in advanced and earlier phases of MM treatment.

## 1. Introduction

Multiple myeloma (MM) is a B-cell malignancy derived from an expansion of clonal plasma cells in the bone marrow, which produce monoclonal immunoglobulins (M-protein) that could settle down into vital organs, causing their progressive dysfunction, with the typical CRAB clinical MM manifestations such as osteolytic bone disease, anemia, or renal failure [1]. MM could be biologically described as a “journey” from a premalignant disorder known as Monoclonal Gammopathy of Undetermined Significance (MGUS), which develops from normal plasma cells via primary genetic events, through the smoldering/asymptomatic MM, until the active MM, which becomes clinically evident. Along this “journey”, secondary genetic events such as genetic abnormalities, epigenetic abnormalities, and cytogenetic abnormalities could take over, resulting in the genetic evolution of the disease and defining the prognosis of MM in terms of its resistance to drugs [2]. The dysregulation of the immune system is an important factor in MM pathogenesis. Thus, trials about drugs acting on the complex relationship between the immune system and the tumor cells could be considered really appealing. As a result, immunotherapy treatment approaches in MM have exponentially evolved over the last two decades. The graft-versus-myeloma effect of allogeneic hematopoietic cell transplantation paved the way for the rising potential role of immunotherapy for MM, having been shown to induce long-term remission and potential cure in a subset of MM patients with a great safety price to pay [3], which has limited its use in real life. First-generation proteasome inhibitors (PIs) and immunomodulatory drugs (IMiDs), bortezomib and thalidomide, respectively, could be considered two of the first available forms of immunotherapy that, replacing standard chemotherapy several years ago, became the backbone of MM treatments with a significant improvement in MM outcomes [4]. The development of second- and third-generation PIs (carfilzomib and ixazomib) and IMIDs (lenalidomide and pomalidomide) could be considered their first evolution, improving their safety profile and strengthening their efficacy [5]. Therefore, monoclonal antibodies (mAb), firstly licensed for RRMM and then also for NDMM settings, have even more improved outcomes in MM patients when administered alone or in association with other drugs [6]. MAbs could represent a basic form of immunotherapy, characterized by the infusion of target-specific antibodies produced from a single clone capable of activating the immune system [7]. These drugs are mostly used in the MM treatment landscape in association, so clinical trials have recently demonstrated the superiority of triplets over duplets [8], quadruplets over triplets [9], and so on in the near future. In the phase 2 MASTER trial [10], 123 NDMM patients received four cycles of daratumumab, carfilzomib, lenalidomide, and dexamethasone (Dara-KRd) as induction and autologous stem cell transplantation (ASCT), followed by 0, 4, and 8 cycles of Dara-KRd consolidation according to MRD status assessed after induction, ASCT, and after the second cycle in each block of Dara-KRd consolidation. Patients with two consecutive MRD negatives discontinued therapy and transitioned to observation and MRD surveillance (MRD-SURE), while patients without two consecutive MRD negatives after consolidation underwent lenalidomide maintenance and MRD assessment after 6 and 18 months. The primary endpoint of the study was reaching MRD of less than 10^−5^ by NGS during treatment, and 81% of patients reached it. Seventy-one percent of patients had two consecutive MRD-negative assessments and entered MRD surveillance, with 52% of them remaining MRD-negative and off therapy [10]. Overall, 2-year PFS was 87% and 2-year OS was 94% [11].

Despite recent advances in outcomes due to the benefits given by novel therapeutic strategies, MM remains incurable, and some disease subgroups continue to not take advantage even of the novel immunotherapies, setting up challenging unmet medical needs. Conventional quadruplets therapies have shown limited benefits for high-risk and ultra-high-risk patients, recently defined as patients harboring 2 or 3 high-risk cytogenetic abnormalities (HRCAs) [12], because novel therapies don’t seem to be able to overcome their negative prognostic impact, even when MRD negativity is obtained. Notably, in the MASTER study, 3-year PFS was 88% in patients with zero HRCAs, 79% in those with one HRCA, and 50% for those with two or more HRCAs after a median follow-up of 42.2 months. With regards to 3-year OS, it was 94%, 92%, and 75%, respectively [10]. Beyond the cytogenetic risk, clinically high-risk disease also negatively impacts outcomes in patients treated with conventional therapies, especially in patients with double-refractory MM. Real life data have demonstrated that outcomes significantly worsen in ≥ triple refractory patients, even in earlier lines of therapy, which is characterized a clinically high-risk feature [13].

In the last few years, novel adoptive immunotherapies have been studied to improve outcomes in triple-refractory patients by targeting other antigens and using novel mechanisms of action. To reduce immune evasion and dysregulation, in multiple hematologic malignancies, bispecific T cell engagers (BiTEs) were developed. These constructs bind a specific tumor antigen as well as CD3 to induce activation of T cells and, consequently, neoplastic cell killing. BiTE antibodies utilize the native T cell repertoire, inducing a polyclonal response with expansion of memory populations [14], differently from chimeric antigen receptor (CAR) T cell therapy. CAR T cells have demonstrated impressive response rates in patients with RRMM and have been approved by the FDA for patients with RRMM disease (idecabtagene vicleucel and ciltacabtagene autoloeucel) [15]. On the basis of exciting results of CART in RRMM, allogeneic CAR T cells are being explored [16], with multiple possible advantages over autologous cells. Allogeneic products could overcome logistical manufacturing challenges by being immediately available as “off-the-shelf” products for therapeutic use in patients with unstable and aggressive disease [17]. Allogeneic CAR T cells could also avoid the challenges of potential RRMM patients’ dysfunctional T cells, having demonstrated increased T cell expansion, cytotoxicity in vitro, and decreased checkpoint marker expression compared with autologous cells [18]. The risks of graft versus host disease (GVHD), the risk of genetic mutations, and the bad expansion and persistence of allogenic CART in vivo remain challenges to overcome in order to endorse this strategy in the MM treatment landscape. An additional promising strategy for MM should be adoptive NK cell therapy, which could have multiple advantages over T cells, such as cytotoxic activity without prior antigen exposure or HLA restriction and a reduced risk of GVHD [19]. 

Despite challenging outcomes, RRMM patients continue to relapse, requiring better study of mechanisms of resistance and evaluating drug combination approaches to overcome them. Given the efficacy of CAR T and BiTEs demonstrated in the setting of advanced disease, moving their employment in the setting of NDMM, especially high- and ultra-high-risk ones, could be a valuable approach to optimizing their role. 

## 2. Mechanisms of Action of Antibody-Drug Conjugates and Bispecific Antibodies

Antibody-drug conjugates (ADCs), which are constructs broadly structured to incorporate an antibody to target the cell of interest, a linker, and a payload, could be considered an emerging method of delivering targeted chemotherapy into neoplastic cells while also eliciting the host immune response against the tumor itself. The cytotoxic payload, mostly anti-microtubule agents, is released into the plasma cell through a linker that can bind the plasma cell itself to a specific receptor, mostly B cell maturation antigen [20]. Through this ligation, ADCs enter the tumor cell and, once internalized and having released their payloads, induce cell cycle arrest, ultimately triggering apoptosis. Suitable payloads should be highly cytotoxic at low concentrations, easily conjugatable to antibodies, and stable when administered in vivo. Anti-mitotic microtubule blockers, such as maytansinoid and auristatin; DNA-damaging agents such as pyrrolobenzodiazepine and calicheamicin; and the RNA polymerase inhibitor such as amanitin, are the most commonly used payloads [21]. Additional anti-tumor mechanisms of ADCs include induction of immunogenic cell death and Fc-mediated effector functions such as antibody-dependent cellular cytotoxicity (ADCC), phagocytosis (ADCP), and complement-dependent cytotoxicity (CDC). In the case of belantamab mafodotin, which will be discussed later in this paper, its prolonged stability in the blood and reduced killing of non-target cells, provided by the uncleavable maleimidocaproyl linker, enhance its efficacy [22]. BCMA, also known as CD269 or Tumor Necrosis Factor receptor superfamily member 17, can be considered one of the most popular targets for ADCs. It is highly expressed on neoplastic plasma cells and late-mature B-cells, being a B-cell activating and myeloma cell survival stimulator factor [23]. BCMA performs these functions via the B-cell activating factor (BAFF) pathway and a proliferation-inducing ligand (APRIL), which are two natural ligands for BCMA itself. Specifically, APRIL binds to BCMA with a significantly higher affinity than BAFF. The consequent activation of BCMA supports the growth and survival of plasma cells via activating MEK/ERK, AKT, NFκB, JNK, p38 kinase, and Elk-1. Membrane BCMA can be cleaved by γ-secretase, resulting in an increase in soluble BCMA. Soluble BCMA can bind to APRIL and BAFF, which may interfere with the BCMA signaling pathway [24]. This is the rationale on which the association of anti-BCMA therapies with γ-secretase is experimented with to improve treatments’ efficacy. 

Modern immunotherapies are based on an old and still valid principle of MM therapy: using T cells as a “weapon against tumor cells”, which has been considered a therapeutic strategy for patients with MM from the allogenic transplant era onward. Furthermore, the availability of lineage-specific markers in MM that can be used to identify plasma cells could be particularly useful to select a specific therapeutic target to hit specific cells. Moreover, these surface proteomes remain stable despite the clonal evolution that is often the basis of disease progression and the development of genetically heterogeneous sub-clones, so these markers could be targeted more often during MM natural history [25]. 

Bispecific T cell antibodies (BsAbs) are modern constructs consisting of two parts, one binding a specific target on plasma cells and the other binding the CD3 receptor on T cells, promoting its activation by the CD3-T cell receptor (TCR) complex. This causes direct T cell activation and subsequent tumor cell killing [26] by a non-chemotherapy approach, differently from ADCs. CD3 engagement induces both the proliferation of CD4 and CD8 T cells, which have cytotoxic activity against the target. The engagement of CD3 is the major proliferation signal, even if additional indirect mechanisms are promoted by the cytokines storm and production of cytolytic molecules, like perforin and granzymes (granzyme B, IFN-γ, IL2, IL6, IL8, IL10, and TNF-α), from T cells, resulting in MM cell lysis. This could be considered an easier mechanism to kill tumors through an autologous patient’s T cell without the need for genetic modifications, as in the more complex CART platforms [27]. The BsAbs-mediated MM cell killing is negatively affected by many cellular and molecular factors, such as bone marrow stromal cells, osteoclasts, regulatory T cells, a proliferation-inducing ligand, transforming growth factor-β, interleukin-6, soluble BCMA, and upregulation in the PD-L1/PD1 axis. Contrariwise, upregulation in effector/target ratio, CD8+ T cells, and differentiated T cells with central and stem-like memory subsets are associated with improved BsAbs-mediated plasma cell lysis. The whole mechanism also involves polyclonal immunological cells, significantly increasing the expression of T cell activation-related parameters and the differentiation of naïve T cells to T cells with memory phenotypes, presumably improving MRD negativity in MM patients [28]. 

BsAbs developed for MM treatment can be divided into two main groups: bispecific T cell engagers (BiTE^®^ (Amgen, Thousand Oaks, CA, USA), consisting of two different single chain variable fragments joined by a flexible peptide linker, and DuoBody^®^ (Genmab A/S, Copenhagen, Denmark), that are monoclonal antibodies with two different antigen-binding fragments (Fab) and a functional constant region fragment (Fc), that could promote the stability of the molecule and increase its half-life [29]. When BiTEs create a synapse with T cells, T cells start to proliferate and increase the effector cells, strengthening the potency of BiTE therapy itself. T cells release perforin and granzyme B towards the tumor cells’ apoptosis [30]. Furthermore, the activation of lymphocytes induces the release of cytokines that amplify the immunological response by involving other immune cells and promoting the proliferation of T cells. BsAbs without Fc lack Fc-mediated effector functions, such as antibody-dependent cell-mediated cytotoxicity (ADCC), antibody-dependent cellular phagocytosis (ADCP), and complement fixation. Preclinical studies have employed Fc mutations in new-generation monoclonal antibodies to modify interaction with FC-γ-receptors and increase or decrease CDC, ADCC, and ADCP [31]. BsAbs are small and highly potent molecules, but they also have a shorter serum half-life, which could not stimulate persistent immunity, necessitating a frequent infusion schedule (weekly or biweekly) and a sine die administration [32]. The first licensed BiTEs were BCMA-directed, which gave evidence of durable responses for some patients, but progression occurred for most patients, necessitating the investigation of possible mechanisms of resistance and finding novel targets to overcome this challenge. Alternative and more recently studied BiTEs were against G-protein coupled receptor family C group 5 member D (GPRC5D), which is highly and heterogeneously expressed on malignant plasma cells and lowly expressed on hair follicles and in the central nervous system (inferior olivary nucleus that relays motor and sensory signals from the spinal cord to the cerebellum regulating motor coordination) [33], independently of BCMA expression [34]. GPRC5D is a good target for novel immunotherapeutic strategies, with preclinical studies showing activity in a BCMA antigen escape model. Moreover, being GPRC5D a GPCR, the exposed epitopes are proximal to the plasma membrane and facilitate tighter immunological T cell-target cell synapses, driving greater cytotoxicity. Unlike BCMA, GPRC5D, a 7-pass transmembrane receptor, cannot be shed into peripheral blood, avoiding a potential mechanism of resistance [35]. It is composed of a Venus Fly Trap domain (VFT), a Cytosine Rich domain, and 7-pass transmembrane helixes, and once the synapses have been created, the immunologic pathway is similar to the other targets. 

Fc receptor-homolog 5 (FcRH5, also known as CD307) is a differentiation antigen type I membrane protein, homologous to the family of Fc receptors, exclusively expressed in the B cell lineage, favoring its safety profile. It is expressed in pre-B cells and on normal plasma cells, being upregulated on malignant plasma cells in MM, mostly where amplification or gain of chromosome 1q21 is present, and maintained on RRMM already treated with proteasome inhibitors or immunomodulatory agents [36]. Its activity has been observed to be correlated with expression levels. Moreover, anti-FcRH5 T cell-dependent bispecific antibodies have demonstrated anti-MM efficacy both in vitro and in murine xenograft models, showing near-complete and highly potent killing of primary MM plasma cells in vitro [36].

## 3. Belantamab Mafodotin

### 3.1. DREAMM Trials 

Belantamab mafodotin is the first-in-class anti-BCMA ADC approved in monotherapy for the treatment of RRMM patients who have received more than four lines of therapy, including an IMID, a PI, and an anti-CD38 antibody, based on the DREAMM-2 phase 2 trial [37]. Belamaf 2.5 mg/kg every 21 days demonstrated a 32% ORR in a heavily pre-treated RRMM population with 7 median prior lines of therapy (range: 6–21), with a median DOR of 12.5 months, a median PFS of 2.8 months, and a median OS of 15.3 months. Median PFS and OS reached 14 and 30.7 months in patients who obtained ≥VGPR, respectively, demonstrating the significant correlation between the depth of response and outcomes. Following updated data from phase 3 of the DREAMM-3 trial, comparing belamaf to the standard of care pomalidomide-dexamethasone, which failed to meet its primary endpoint of PFS benefit, the drug was withdrawn from the USA market. The DREAMM-3 trial showed a median PFS of 11.2 months compared with 7.0 months for belamaf vs. pomalidomide plus dexamethasone, respectively, at a median follow-up of 11.5 months for the belantamab mafodotin arm and 10.8 months for the comparator arm [38].

In order to improve belamaf efficacy, the DREAMM clinical development program is expanding its clinical trials offer, both experimenting with belamaf in association with other drugs with different target/mechanisms of action and placing it earlier in the therapeutic history of MM patients. Phase 1/2 DREAMM-4 clinical trial enrolled RRMM patients who have received ≥3 previous lines of therapy, including a PI, an IMID, and an anti-CD38 mAb, and explored different combinations in seven sub-studies of part 2, whose primary endpoint was ORR. Belamaf was experimented in combination with GSK3174998 (anti-OX40 monoclonal antibody), feladilimab (inducible T cell costimulatory agonist), nirogacestat (gamma secretase inhibitor), dostarlimab (anti-PD1 monoclonal antibody), isatuximab (anti-CD38 monoclonal antibody), nirogacestat plus lenalidomide and dexamethasone, nirogacestat plus pomalidomide and dexamethasone, in each of seven different sub-studies, respectively. This trial is actually recruiting patients, and is estimated to be completed February 2028 (NCT03848845). The first interim analysis of DREAMM-4, presented at the 2022 EHA Meeting, was related to the combination of belamaf plus pembrolizumab, a PD-1 checkpoint blockade antibody whose previous studies in MM failed their primary endpoints, discouraging the experimentation of IMIDs plus anti-PD-1 because of their dismal safety profile. Part 1 established the dose of belamaf 2.5 mg/kg in association with pembrolizumab 200 mg, both IV Q3W up to 35 cycles, for the Part 2 expansion. Data from 34 patients (6 in Part 1 and 28 in Part 2) with 5 median prior lines of therapy (range 3–13) showed 47% ORR with most responses (10/16 pts) ≥ VGPR. At the median follow-up of 14.7 months, the median DOR was 8.0 months, and the median PFS was 3.4 months. The most common (≥35%) adverse events were keratopathy (any grade: 76%; grade ≥ 3: 38%), blurred vision (grade ≥ 3: 0%), and thrombocytopenia (any grade: 35%; grade ≥ 3: 29%). Dose discontinuations were not reported [39]. Data from 34 patients treated with low-dose belamaf plus nirogacestat and 37 patients with belamaf monotherapy in the phase 1/2 DREAMM-5 platform sub-study 3 were presented at the last 2023 EHA Meeting, demonstrating encouraging 29% and 38% ORR in combination and monotherapy arms, respectively. Nirogacestat is a gamma secretase inhibitor that prevents the cleavage of several transmembrane proteins, including BCMA, preventing its shedding into peripheral blood. The authors reported a substantial reduction of high-grade ocular events, with grade 3 ocular events less frequent for low-dose belamaf plus nirogacestat (29% vs. 59%), highlighting an increase in BCMA target density with nirogacestat [40]. DREAMM-6, phase 1/2, open label, dose escalation, and expansion study evaluated the safety, tolerability, and clinical activity of belamaf in combination with lenalidomide plus dexamethasone (Arm A) or bortezomib plus dexamethasone (Arm B) in RRMM patients with ≥1 prior lines of therapy. Patients in Arm A received 4 belamaf doses/schedules (1.9 mg/kg Q8W or Q4W; 2.5 mg/kg Q4W or Q4W SPLIT dose [50% on Days (D)1, 8] IV) in combination with lenalidomide (20 mg PO D1–21) and dexamethasone (20 mg PO/IV D1, 8, 15, 22). Data from the interim analysis were presented at the 2022 EHA Meeting, reporting a median DOR of 11.1 months, only reached in the 1.9 mg/kg Q4W cohort. At the time of the data cut (23 July 2021), the median PFS was not reached. As for safety, grade ≥3 keratopathy occurred in 0 patients in 1.9 mg/kg Q8W, 1 patient (25%) in 1.9 mg/kg Q4W, 8 patients (50%) in 2.5 mg/kg Q4W, and 6 patients (46%) in 2.5 mg/kg Q4W SPLIT cohorts, demonstrating a safe profile without new safety concerns [41]. This trial is currently active, not recruiting, with an estimated study completion date of February 2024 (NCT03544281). Phase 3 of the DREAMM-7 trial compared belamaf in combination with bortezomib and dexamethasone with daratumumab plus bortezomib and dexamethasone in RRMM with ≥1 prior lines of therapy. This trial is currently active, not recruiting, with an estimated study completion date of June 2026 (NCT04246047). Phase 3 of the DREAMM-8 trial employed belantamab plus pomalidomide and dexamethasone vs. pomalidomide plus bortezomib and dexamethasone in RRMM patients with ≥1 prior lines of therapy. It is currently recruiting, with an estimated study completion date of December 2027 (NCT04484623). 

For transplant-ineligible NDMM patients, belamaf was experimented with in combination with standard of care in the DREAMM-9 phase 1 trial. The study evaluated different doses/dose schedules of belamaf in combination with bortezomib, lenalidomide, and dexamethasone in up to eight cohorts and will determine the recommended Phase 3 dose (RP3D) (NCT04091126). The interim analysis presented at the 2023 ASCO Meeting showed data from 93 patients treated across cohorts 1–7, confirming no new safety signals and providing early and deep anti-myeloma responses with high MRD negativity rates. The cohort 1, which was the one with the longest follow-up of 27.6 months, demonstrated a 100% ORR with 75% MRD negativity among patients having obtained ≥CR, with belamaf at the dose of 1.9 mg/kg Q3/4W [42]. Phase 1 DREAMM-20, an open-label, multicenter, dose-escalation, and expansion study, aimed to investigate the safety, tolerability, and clinical activity of belantamab as monotherapy and in combination with other treatments (back bone: lenalidomide plus dexamethasone) in participants with RRMM who have received ≥1 prior lines of therapy (Part 2). Part 3 will enroll transplant-ineligible NDMM patients. The trial is not yet recruiting, and the completion date is estimated to be June 2027 (NCT05714839). 

Preclinical data demonstrated significantly increased direct and indirect anti-MM activity for belamaf in combination with other therapies with different mechanisms of action. This suggests these combinations could improve the clinical efficacy already seen with belamaf monotherapy. This requires platform trials, which, differently from usual trials, offer the advantage of evaluating multiple interventions concurrently as well as sequentially, where the multiple therapies are experimental and/or already licensed [43]. This type of trial design can also evaluate biomarkers of interest, paving the way for personalized medicine.

### 3.2. ALGONQUIN Study

The multicenter ALGONQUIN phase I study [44], conducted by the Canadian Myeloma Research Group, explored different doses and schedules of belamaf combined with pomalidomide and dexamethasone (Bela-Pd) in RRMM patients who had received ≥1 prior line of therapy and were exposed to lenalidomide, PIs, but pomalidomide naïve. In Part 1 of the study, a dose-escalation phase, patients received pomalidomide 4 mg days 1–21, dexamethasone 40 mg weekly (20 mg in patients older than 75 years), and belamaf single (1.95 or 2.5 mg/kg day 1) or split (2.5 or 3.4 mg/kg spit on days 1 and 8) every 4 weeks. The primary endpoint of this part was to identify the MTD and RP2D to use in Part 2, aiming to evaluate ORR. Fifty-four patients, with a median age of 67.5 years, were enrolled. The median number of prior lines of therapy was 3 (range 2–5), and 72.2% were triple-refractory. Across all cohorts, ORR was 86%, with 60% of patients achieving at least a VGPR. After a median follow-up of 5.7 months, the median PFS resulted in 15.6 months. The most common grade ≥ 3 adverse events included keratopathy (55%), neutropenia (37%), thrombocytopenia (27.5%), and decreased BCVA (23.5%). 

## 4. Teclistamab

### 4.1. MajesTEC-1 Trial

Teclistamab was approved by the FDA in 2022, at a dose of 1.5 mg/kg SC QW, as the first humanized IgG Fc BCMA-directed CD3 T cell engager to be indicated for the treatment of RRMM patients who have received at least 4 prior lines of therapy (at least 3 prior therapies in EU), including a PI, an IMID, and an anti-CD38 mAb, on phase 1/2 MajesTEC-1 trial results [45]. After a median follow up of 23 months, ORR was 63% with 59.4% deep responses ≥VGPR, median DOR was 21.6 months, and 81% of patients were MRD negative at 10^−5^ at day 100. Patients have received a median of 5 prior lines of therapy (range 2–14), and patients who have received ≤3 previous lines obtained a higher ORR (74.4%) than more pre-treated patients (59%). Median PFS and median OS were 11 and 22 months, respectively, rising in patients with ≥CR [45]. At the data cutoff, 47 out of 165 patients remain on treatment. 

### 4.2. Teclistamab-Based Combinations: MajesTEC-2 Study

Even if BsAbs have given great outcomes in very challenging patients’ populations, RRMM patients relapsed, and resistance mechanisms need to be more precisely understood. Considering that effector T cell function is often compromised in MM, with progressive dysfunction observed over disease progression and advanced treatment lines [46], clinical trials have been built to improve efficacy by moving up immunotherapies in earlier lines of therapy. Furthermore, combining them with other target drugs could help overcome mechanisms of resistance and improve outcomes, even in advanced lines of therapy. The MajesTEC-2 phase 1b multicohort trial [47] experimented with the combination of teclistamab and lenalidomide in RRMM patients who have received ≥2 prior lines of therapy, including a PI, an IMiD, and an anti-CD38 antibody. At a median follow up of 9.9 months, ORR was 74.2% (35.5% of them were ≥CR). Among responders, at a median follow-up of 10.1 months, the median time to first response was 1.2 months, and the median duration of response was not reached. These preliminary data confirm the efficacy of the doublet containing lenalidomide in a pre-treated RRMM population, 41.9% of whom were lenalidomide-refractory, without new cumulative safety concerns, supporting the potential of this combination [47]. Recent data from the 2023 EHA Meeting described the teclistamab plus nirogacestat cohort of the same trial, which enrolled RRMM patients with ≥3 prior lines of therapy or double refractory (defined as refractory to a PI and an IMiD) and triple exposed (defined as exposed to a PI, an IMiD, and an anti-CD38 antibody) with progressive disease within 12 months of their last line of therapy. Patients had 4 median prior lines (range 2–12); the best 92.3% ORR was obtained with the teclistamab dosing of 1500 µg/kg QW plus QD delayed dose of nirogacestat, being all deep responses ≥VGPR. ORR were 57–92% among the three dose levels assessed, providing insights on the combination of BCMA-directed therapies and gamma secretase inhibitors [48]. Other ongoing teclistamab-based combinations trials are described in Table 1.

## 5. Elranatamab 

### MagnetisMM-1 and MagnetisMM-3 Trials

Elranatanab is an investigational, off-the-shelf, humanized BCMA-CD3-directed BsAb whose arm binds directly to specific antigens on cancer cells and the other arm binds to T cells, bringing both cell types together. The binding affinity of elranatamab for BCMA and CD3 has been engineered to elicit potent T cell-mediated anti-myeloma activity. Elranatamab is under review by regulatory agencies. The Food and Drug Administration (FDA) has granted priority review for its Biologics License Application (BLA) in RRMM. The European Medicines Agency (EMA) is reviewing a marketing authorization application (MAA) for elranatamab under the PRIME scheme. Elranatanab demonstrated great results in the phase 1 MagnetisMM-1 trial [49] that enrolled patients with progression on standard therapies including at least one PI, one IMiD, and one anti-CD38 mAb. Prior BCMA-targeted therapy was permitted. Fifty-five RRMM patients with five median prior lines of therapy (range: 2–14), 91% triple-refractory, and 23.6% with prior BCMA-targeted therapy received subcutaneous elranatamab monotherapy. ORR was 64% overall and 54% among patients who have already received an anti-BCMA treatment, with 56% of patients achieving at least VGPR and 38% CR or better. At a median follow up of 12 months, the median DoR was 17.1 months, the median PFS was 11.8 months, and the median OS was 21.2 months. All 13 patients with at least CR and MRD evaluable, achieved MRD negativity, which was maintained for more than 1 year in almost one-third, showing deep responses with elranatamab monotherapy [49]. The ongoing phase 2, open-label, multicenter, and non-randomized MagnetisMM-3 study [50] experimented with elranatamab monotherapy in 187 RRMM participants who were refractory to at least one PI, one IMiD, and one anti-CD38 monoclonal antibody and had to be relapsed or refractory to their antimyeloma treatment. Cohort A enrolled patients without prior anti-BCMA therapy, while patients who have already received anti-BCMA ADC or CART were enrolled in cohort B. All the participants received elranatanab at a dose of 76 mg SC QW on a 28-day cycle, with the ORR being the primary endpoint. Cohort A enrolled 123 patients with a median age of 68 years (36–89), 5 median prior lines of therapy (range: 2–22), triple-class refractoriness in 100% of patients, and penta-drug refractoriness in 70.7%. ORR was 61%, including a ≥VGPR rate of 56%, and the MRD negativity rate at a 10^−5^ sensitivity was 89.7% in patients who achieved ≥CR. At a median follow-up of 14.7 months, the median PFS and OS had not yet been reached, being the 15-month PFS and OS rates of 50.9% and 56.7%, respectively [50]. Elranatamab has also demonstrated efficacy and a manageable safety profile in elderly (<65 vs. ≥65 years) or frail (frailty according to the simplified frailty score) patients with RRMM in a sub analysis recently presented at the last ASCO Meeting [51]. Particularly, the discontinuation rate occurred in 62.8% vs. 67.5% of patients aged <65 vs. ≥65 years and in 63.1% vs. 71.8% of the non-frail vs. frail groups, respectively, suggesting a similar safety profile between the two groups. 

In a recent retrospective study comparing data from MagnetisMM-3 patients vs. two US-based oncology electronic health record databases including triple refractory RRMM patients, a significant benefit in ORR for patients treated with elranatamab in the trial rather than in real life treatments was found [52]. Also in the Netherlands, an unanchored MAIC among elranatamab vs. belamaf vs. selinexor-dexamethasone vs. standard therapies demonstrated a clear clinical benefit of elranatamab for triple refractory RRMM patients who achieved a significantly higher ORR than that obtained with the other considered regimens [53]. Other ongoing elranatamab-based combinations trials are described in Table 2.

## 6. Linvoseltamab: LINKER-MM Trials

Linvoseltamab (REGN5458) is a more recently studied BCMA × CD3 BsAb, which demonstrated 64% (n = 58) and 50% (n = 104) ORR in the 200 mg and 50 mg cohorts, respectively, at a median follow up of 2.3 and 4.7 months in the phase 1/2 LINKER-MM1 trial [54]. Furthermore, the 200 mg dose seemed to have consistent efficacy across high-risk subgroups and induce responses in patients who have progressed on 50 mg dosing. These data have encouraged an active program of compassionate use of linvoseltamab (NCT05164250). Based on these efficacy data, the phase 1/2 LINKER-MM4 trial aims to employ linvoseltamab in the NDMM population, but this trial is not enrolling patients yet. Phase 3 of the randomized LINKER-MM3 trial, not yet recruiting patients, aims to compare linvoseltamab to the standard of care elotuzumab-pomalidomide-dexamethasone in RRMM patients who have received from 1 to 4 previous lines of therapy. The only ongoing trial that employs linvoseltamab in combination with other drugs in RRMM is the phase 1b NCT05137054. It is composed of nine different cohorts that combine linvoseltamab with different drugs such as daratumumab, carfilzomib, lenalidomide, bortezomib, pomalidomide, isatuximab, finlimab, cemiplimab, and nirogacestat separately, with the primary endpoint of dose finding and safety. 

## 7. Talquetamab

### 7.1. MonumenTAL-1 Trial

Talquetamab represents the first-in-class BsAb targeting GPRC5D (G protein-coupled receptor of family C, group 5, member D) to have been explored in MM, starting from preclinical studies and subsequently in clinical trials. After flow cytometry showed higher GPRC5D levels on MM cells compared with normal plasma cells, talquetamab was found to kill MM cells obtained from bone marrow aspirates of NDMM and double- or triple-class refractory MM patients by redirecting T cells against GPRC5D antigen on the MM cell surface [55]. The multicenter phase 1/2 MonumennTAL-1 [56] has been the first trial to evaluate talquetamab in RRMM patients with a median of 6 previous lines of therapy and triple- and penta-drug-refractoriness in 79% and 30% of them, respectively. Part 1 of the study, a dose-escalation phase aiming to assess the safety profile of talquetamab, included 232 MM patients with disease progression on or intolerant of all established therapies. They received talquetamab intravenously (102) or subcutaneously (130) at different doses and schedules, and, since the maximum tolerated dose was not reached, the two subcutaneous doses of 405 µg/kg administered weekly (with step-up doses of 10 and 60 µg/kg) and 800 µg/kg given every other week (with step-up doses of 10, 60, and 300 µg/kg) were chosen for Part 2 of the trial, based on results from Part 1. As per the safety profile, cytokine release syndrome (CRS) occurred in 77% and 80% of patients receiving 405 µg/kg and 800 µg/kg, respectively, being of grade 1–2 in most patients. CRS started a median of 2 days after treatment, and the median duration was 2 days for both doses. Among hematologic side effects, considering all subcutaneous cohorts, 45.4% developed grade 3–4 neutropenia and 20% developed grade 3–4 thrombocytopenia. Forty-seven percent and 34% of patients receiving talquetamab 405 µg/kg and 800 µg/kg, respectively, experienced any grade infections, being 7% of grade 3–4 for both doses. Neurologic events, mainly confusion state and anosmia, developed in 7.7% of patients and were grade 1 or 2. Peculiar toxicities of subcutaneous talquetamab were found to be skin-related side effects, including exfoliation, pruritus, and dry skin, occurring in 60% of patients (grade 3–4 = 0.8%) and lasting a median of 39 days; dysgeusia, documented in 56% of patients; and nail toxic effects, including disorders of the nail bed, nails coming away from the nail bed, changes in nail color, shape, texture, and growth, shedding of the nail, and nail ridging, occurring in 39% of patients. Regarding activity, ORR was 70% (at least VGPR = 57%) at the 405 µg/kg dose and 64% (at least VGPR = 52%) at the 800 µg/kg dose, with a median duration of response of 10.2 months and 7.8 months, respectively. 

Phase 2 of the MonumenTAL-1 trial [57] enrolled RRMM patients who had received at least three prior lines of therapy, including at least one IMiD, one PI, and one anti-CD38 mAb. Overall, considering all patients in both Phase 1 and Phase 2 of the trial, 143 patients received talquetamab at 0.4 mg/kg weekly, 143 received 0.8 mg/kg every other week, and 51 patients with prior anti-BCMA CAR T cells or BsAb were treated with either dose. In the 0.4 mg/kg cohort, triple- and penta-class refractory patients were 74% and 29%, respectively, whereas in the 0.8 mg/kg cohort they were 69% and 23%. ORR, the primary endpoint of the phase 2 study, was 74.1% in the 0.4 mg/kg cohort and 71.7% in the 0.8 mg/kg cohort, with 59.4% and 60.7% of patients achieving at least VGPR, respectively. Median PFS was 7.5 months in the first cohort and 14.2 months in the latter cohort, whereas OS at 12 months was 76.4% and 77.4%, respectively. Notably, among 51 patients with prior anti-BCMA therapies, ORR was 64.7%, 54.9% of them obtained at least VGPR, and the median PFS was 5.1 months. In the whole study population, any grade CRS occurred in 76.7% of patients, resulting in a grade ≥ 3 in only 1.4% of them. Skin- and nail-related side effects were mostly grade 1–2 and occurred in 65% and 55% of patients, respectively, whereas dysgeusia developed in 72% of them. The most frequent grade 3–4 side effects were anemia, neutropenia, and thrombocytopenia, occurring in 29%, 30%, and 21% of patients who, in 64% of cases, experienced any grade of infection. More specific details about infections will be reported in a later chapter. 

### 7.2. Talquetamab-Based Combinations: TRIMM-2 Study

Phase 1 of the TRIMM-2 clinical trial (NCT04108195) is evaluating talquetamab in combination with daratumumab with or without pomalidomide with the aim of identifying the recommended dose for each treatment to use in part 2 of the study, exploring the safety and activity of combinations. In the last EHA meeting, data on the combination talquetamab plus daratumumab was presented [58]. The rationale for this combination is based on results from preclinical studies showing that depletion of CD38-expressing Tregs by daratumumab enhances talquetamab-mediated killing of MM cells [55]. Moreover, reduced NADase activity of CD38 by daratumumab increases the antitumor T cell immune response in mouse models [59]. The TRIMM-2 study enrolled patients who had received at least three prior lines of therapy or were refractory to a PI and IMiD. Prior therapy with anti-CD38 mAb for > 90 days was allowed, as was prior immunotherapy with BsAbs or CAR T cells. Patients received subcutaneous talquetamab 0.4 mg/kg weekly (14 patients) or 0.8 mg/kg (51 patients) every other week with step-up dosing in combination with subcutaneous daratumumab 1800 mg per approved schedule. Overall, median age was 63 years, median prior lines of therapy was 5 (2–16), and 60% and 28% of patients were triple- and penta-refractory, respectively. Moreover, 20% of patients have been exposed to bispecific antibodies and 17% to CAR T cell therapy, in both cases targeting BCMA. CRS occurred in 78.5% of patients, and, despite all grades 1–2, 43% of patients received tocilizumab as a supportive measure; only 4.6% of patients developed grade 1–2 ICANS. In regard to the peculiar toxicities of talquetamab, oral adverse events, mainly dysgeusia, dry mouth, and stomatitis, occurred in 89% of patients, being of grade 3–4 in 3% of them; skin toxicities and nail adverse events were documented in 81.5% (grade 3–4 = 9%) and 66% of patients (grade 3–4 = 1.5%), respectively. The most frequent grade 3–4 hematologic toxicities were neutropenia (27%), lymphopenia (26%), and thrombocytopenia (21.5%), whereas pneumonias represented the most frequent infections, occurring in 26% of patients (12% grade 3–4). Considering the two cohorts of patients, ORR was 71.4% and 84% in the 0.4 mg/kg and 0.8 mg/kg groups, respectively, with 57% and 74% achieving at least VGPR, respectively. Very promising results were obtained with the 0.4 mg/kg every other week schedule in patients refractory to anti-CD38 mAb in whom ORR was 80% and in those exposed to CAR T cell therapy or bispecific achieving an ORR of 79%. Overall, median PFS was 19.4 months, and OS at 12 months was 92% [58]. The ongoing phase III MonumenTAL-3 trial (NCT05455320) is comparing subcutaneous talquetamab with daratumumab and pomalidomide (Tal-DP) with talquetamab and daratumumab (Tal-D) vs. daratumumab, pomalidomide, dexamethasone (DPd) in RRMM patients who have received at least 1 prior line of therapy.

### 7.3. Talquetamab-Based Combinations: The RedirecTT-1 Study

As above described, teclistamab represents the first BCMA-targeting BsAb to have been approved for the treatment of triple-class exposed RRMM patients based on the results of the MajesTEC-1 study. Therefore, since they target different antigens on MM cells, it seemed exciting to explore the combination of them that the Phase 1b/2 RedirecTT-1 study (NCT04586426) is evaluating in RRMM patients who have been exposed to a PI, IMiD, and anti-CD38 mAb [60]. Primary endpoints of the study are the safety of the combination and the identification of a recommended Phase 2 regimen (RP2R). Ninety-three patients with a median age of 65 years (range 39–81) received all doses of teclistamab plus talquetamab; the median prior lines of therapy were 4 (1–11); 79.6% and 25.8% of patients were triple- and penta-class refractory, respectively. Moreover, 37.6% of patients had extramedullary plasmacytomas. The most frequent adverse event was CRS, occurring in 76.3% of patients and being of grade 3 in 3%. It mostly developed during step-up dosing, or cycle 1, and had a median duration of 2 days. Dysgeusia, skin toxicity, and nail disorders were documented in 61.3%, 53.8%, and 46.2% of patients, respectively (no grade 3–4), whereas, among hematologic toxicities, grade 3–4 neutropenia and thrombocytopenia occurred in 61.3% and 29% of patients, respectively. Thirteen percent of patients experienced febrile neutropenia, and although pneumonia occurred in 26.9% of all patients, this rate was found to be 11.8% in patients treated with RP2R consisting of telistamab 3 mg/kg and talquetamab 0.8 mg/kg, both every other week. However, discontinuation due to drug-related adverse events was 6.5%. As per efficacy, ORR was 86.6% (at least VGPR = 40.2%) across all dose levels and 96.3% (at least VGPR = 40.7%) at the RP2R. Overall, median DoR was not reached after a median follow-up of 13.4 months, and 9-mo PFS was 70% (77% at RP2R). Notably, at RP2R, 85.7% of patients with extramedullary disease had ORR, with 28.6% achieving at least CR.

In the Table 3, we summarized other ongoing trials exploring talquetamab-based combinations.

## 8. Cevostamab

Cevostamab represents the first BsAb targeting FcRH5 on myeloma cells and CD3 on T cells under evaluation in RRMM. In the ongoing Phase I GO39775 study [61], 161 RRMM patients with no available therapies received intravenous cevostamab every 3 weeks with step-up dosing in cycle 1, for 17 cycles. The median age was 64 years (range 33–82), and the median prior lines of therapy were 6 (2–18). Most patients were triple-class refractory (84.5%), 68.3% were penta-drug refractory, and 33.5% had received at least one prior anti-BCMA targeting agent. The most common side effect was CRS, which occurred in 80.7% of patients (only 1.2% in grade 3), was primarily observed in cycle 1, and resolved within 48 h of onset in 85% of patients. Immune effector cell-associated neurotoxicity syndrome (ICANS), mainly grade 1–2, occurred in 14.3% of patients. Neutropenia and thrombocytopenia were the most frequent grade 3–4 hematologic toxicities, whereas infections developed in about 45% of patients. ORR was found to increase with target dose, and patients receiving a dose level of 132–198 mg had an ORR of 56.7% (≥VGPR = 33.3%). The median duration of response was 11.5 months after a median follow-up of 14.3 months. At the last ASH meeting, data regarding 18 patients who completed 17 cycles was presented [62]. A VGPR or better was obtained in 17/18 patients (94%) by the time of completion of therapy, with 61% achieving at least a CR. Moreover, 7/18 patients remained in response ≥12 months after therapy conclusion. Notably, retreatment with cevostamab was able to induce a durable response in a not negligible portion of patients. Other ongoing cevostamab-based combinations trials are described in Table 4.

## 9. Peculiar Toxicities of Belantamab Mafodotin and Bispecific Antibodies

Ocular toxicity represents the most relevant adverse event of belantamab mafodotin, as documented in DREAMM-1 and DEAMM-2 clinical trials. It includes several clinical pictures as keratopathy, characterized by corneal epithelial changes named MECs (microcystic-like epithelial changes) documented by slit-lamp examination, with or without ocular symptoms; best-corrected visual acuity (BCVA) changes; blurred vision; and dry eye. It is thought that belamaf enters the cornea and, after it is internalized by corneal epithelial cells through micropinocytosis, induces apoptosis of these cells. Changes in vision occurr when corneal epithelial cells containing belamaf that have not yet undergone apoptosis migrate near the visual axis [63]. In the pivotal DREAMM-2 trial, grade 1–2 keratopathy occurred in 43% and 54% of patients receiving belamaf 2.5 mg/kg and 3.4 mg/kg, respectively, whereas grade 3 keratopathy was seen in 27% and 20% of the first and latter cohorts, with only 1% of patients developing grade 4 keratopathy in the 3.4 mg/kg group [64]. BCVA reduced to 20/50 or worse was documented in 48% and 49% of the 2.5 mg/kg and 3.4 mg/kg cohorts, respectively, whereas blurred vision occurred in 25% and 36% and dry eye in 18% and 25%, respectively [65]. Ocular events resolved in 36% of patients receiving 2.5 mg/kg and in 28% of those treated with 3.4 mg/kg, with a median time to resolution of 71 days and 96 days, respectively. To minimize the risk of ocular toxicity, it is imperative to conduct a baseline ophthalmic assessment prior to the first dose of belamaf and prior to each subsequent dose to monitor the appearance of eye symptoms. Moreover, patient education plays a critical role since the rapid identification of ocular events can decrease treatment discontinuations. 

Cytokine release syndrome is the most frequent non-hematologic adverse event occurring in patients receiving BsAbs, as pictured in Table 5. CRS represents a systemic inflammatory response occurring when binding of BsAb on plasma cells and effector cells leads to the release of cytokines such as IFN-gamma or TNF-alpha, resulting in the further activation of immune and non-immune cells to produce pro-inflammatory cytokines [66]. Notably, IL-6 plays a crucial role in CRS, and its concentration correlates with the severity of CRS [67]. CRS can range from mild symptoms such as fever, myalgia, arthralgia, and anorexia to more severe and life-threatening situations such as hemodynamic instability and organ dysfunction [68]. Recent studies suggest that prophylactic tocilizumab prior to bispecific antibody administration can reduce the incidence of CRS. The first prospective study evaluating this issue included 26 patients enrolled in the MajesTEC 1 trial who received a single dose of tocilizumab (8 mg/kg intravenously) ≤ 4 h before the first teclistamab step-up dose. Prophylactic tocilizumab reduced the incidence of CRS to 26% (all grades 1–2) with no impact on response to teclistamab [69]. Also, in the ongoing GO39775 phase I study, a group of patients received a single 8 mg/kg dose of tocilizumab 2 h before the initiation of intravenous cevostamab. The rate of CRS was significantly lower in patients receiving a pre-treatment with tocilizumab since all grade CRS occurred in 90.9% (grade 1–2: 88.6%) vs. 38.7% (grade 1–2: 35.5%) in the not-tocilizumab and tocilizumab pre-treatment groups, respectively. As with teclistamab, prophylactic tocilizumab did not impact the anti-myeloma activity of cevostamab [70].

The increased risk of infection in patients receiving BsAbs is related to several factors, including prolonged severe hypogammaglobulinemia caused by plasma cell aplasia, prolonged neutropenia and lymphopenia, as well as T cell exhaustion [71]. Moreover, patients who can receive BsAbs may have pre-existing immunosuppression related to advanced disease and to prior therapies such as anti-CD38 mAbs. A retrospective study of 39 patients who received BsAbs between 2018 and 2022 in a single Australian center [72] showed that 90% of patients developed at least one episode of infection. Overall, among 111 infection episodes occurring at a median of two BsAbs cycles, 30% were microbiologically documented and caused by viruses (58%), bacteria (39%), and acid-fast bacilli (3%). In 57% of episodes, hospital admission was required, with a median length of stay of 4 days (range 3–7).

Another recent retrospective analysis aimed to evaluate infectious complications and factors affecting them in 96 patients treated with BCMA or GPRC5D BsAbs on early-phase clinical trials at three academic institutions in the US [73]. The infection rate per 100 days was 0.57 and 0.13 for BCMA and GPRC5D BsAbs monotherapy, with the rate of grade ≥ 3 infections higher in the BCMA group vs. GPRC5D group (58% vs. 36%, *p* = 0.04). Eight percent of patients, all receiving BCMA BsAbs, died of infections. The incidence of any-grade infections at 9 months was significantly higher in the BCMA vs. GPRC5D monotherapy group (57% vs. 16%, *p* = 0.012). Multivariate analysis found BCMA BsAbs, history of previous infection, baseline lymphopenia, and baseline hypogammaglobulinemia as factors significantly associated with a higher risk of grade ≥ 3 infections. Although monthly IVIG replacement in patients with IgG levels < 400 mg/dl is recommended, prospective studies aiming to clarify the optimal duration and dosing of IVIG are awaited. Pneumonia represents the most common clinically documented infection, and it occurred in 18% of patients treated with teclistamab [74] and in 16.3% of those receiving elranatamab [75], so patients receiving BsAbs should be closely monitored for early diagnosis and treatment of these infections.

Besides hypogammaglobulinemia, persistent B cell and plasma cell suppression are associated with a high risk of viral infections that can be caused by CMV, EBV, parvovirus, HSV, BK polyomavirus, and other visceral infections [76]. Moreover, these patients show a particularly high risk of SARS-CoV-2 infections, and any grade COVID-19 occurred in 29.1% (grade 3–4: 21.2%) and 29.3% (grade 3–4: 15.4%) of patients treated with teclistamab and elranatamab, respectively [45,75].

## 10. Treatment with Sequential BCMA-Targeted Therapies and Real Life Experiences

Among the different available BCMA immunotherapies it is difficult to establish the best timeline to use them due to the lack of data about their sequence in MM. Real world studies have demonstrated worse outcomes in patients treated with BCMA-directed CAR T cells after another BCMA-targeted immunotherapy when compared to BCMA-naïve patients, mostly if patients have already received BCMA-directed bispecific antibodies, with a median PFS lower than 3 months [77,78]. Factors affecting worse responses were identified as the duration of the last BCMA-directed therapy, the time between previous therapy and lymphocyte apheresis or infusion, and the penta-refractoriness of patients [79,80]. Based on the principle that BCMA target loss is a rare event that mediates MM resistance to BCMA-targeted immunotherapies, recent data demonstrated novel BCMA mutations as possible resistance mechanisms that negate BCMA-directed drug activity, despite detectable surface BCMA protein expression [81].

Real world data about BCMA-directed bispecific antibodies are recently emerging and confirm their efficacy in a MM population, despite the fact that 80% of patients were considered ineligible for trial inclusion, as Midha et al. presented at the 2023 International Myeloma Society Annual Meeting [82]. Similarly, real world USA experiences about ide-cel confirmed good ORR and outcomes in populations not always eligible in KARMMA trials [83,84]. Real life experiences are really important to understand the applicability of these novel therapies in real world settings, where triple-refractory patients could be more pre-treated, less fit, or older than those enrolled in clinical trials. Data from clinical trials are hard to compare with those from real world studies due to differences between study populations, but they could be useful to focus on specific settings where novel drug employment could be developed. 

## 11. Conclusions

Novel immunotherapies such as ADC and BsAbs showed undoubted clinical efficacy as monotherapy in patients with very advanced and multi-refractory MM, with a manageable safety profile. Based on the actual inability to reach a “cure” in MM, ongoing clinical studies are evaluating BsAbs in combination with other agents as IMiDs and PIs anti-CD38 mAbs, from which we could expect a further increase in their anti-myeloma activity, with the interesting question of novel toxicities to fight. Moreover, clinical trials exploring the combination of BsAbs targeting different antigens on plasma cells are very interesting, foreshadowing the possibility of pure immunotherapy for MM, deleting the standard therapy toxicity. The ongoing and future trials will have to fill an increasingly pressing clinical need, represented by the growing MM population becoming multi-refractory early; therefore, new immunotherapeutic combination strategies are expected to prolong the survival of these patients, who previously had a very poor outcome. On the other hand, by anticipating novel treatment strategies in the earliest lines of therapy, including induction, we hope that survival measures, already significantly improved with quadruplet combinations, can be further lengthened. New peculiar toxicities have emerged, but the increasing use of BsAbs is making the management of adverse events, such as CRS and infections, easier. However, despite our expectations, several factors, such as high costs for bispecific antibodies, manufacturing limitations, and complex logistics for the case of CAR T cells, could make these revolutionary therapies accessible to a limited proportion of patients with ethical implications. The effort will be to identify the best therapy for each patient in order to achieve the best results without wasting resources.

## Figures and Tables

**Table 1 pharmaceuticals-16-01628-t001:** Teclistamab-based combination trials.

Trial	Phase	Population	Intervention	Trial ID
MajesTEC-4	III	NDMM maintenance	Teclistamab vs. lenalidomide monotherapy	NCT05243797
MajesTEC-3	III	RRMM with ≤3 prior lines of therapy	Teclistamab, daratumumab sc vs. daratumumab sc, pomalidomide, and dexamethasone or daratumumab sc, bortezomib, and dexamethasone	NCT05083169
MajesTEC-7	III	NDMM	Teclistamab, daratumumab, lenalidomide (Tec-DR) and talquetamab, daratumumab, lenalidomide (Tal-DR) vs. daratumumab, lenalidomide and Dexamethasone (DRd)	NCT05552222
MajesTEC-9	III	NDMM	Teclistamab monotherapy vs. teclistamab with pomalidomide plus bortezomib and dexamethasone or carfilzomib plus dexamethasone	NCT05572515

**Table 2 pharmaceuticals-16-01628-t002:** Elranatamab-based combination trials.

Trial	Phase	Population	Intervention	Trial ID
MagnetisMM-4	IB/II	RRMM with ≥3 prior line	Elranatamab + Nirogacestat Elranatamab + lenalidomide + dexamethasone	NCT05090566
MagnetisMM-3	II	RRMM	Elranatamab monotherapy	NCT04649359
MagnetisMM-20	IB	RRMM	Elranatamab Carfilzomib Maplirpacept	NCT05675449
MagnetisMM-5	III	RRMM	Elranatamab Daratumumab Pomalidomide Dexamethasone	NCT05020236
MagnetisMM-6	III	RRMM	Elranatamab Daratumumab Lenalidomide Dexamethasone	NCT05623020
A Study Evaluating the Safety, Pharmacokinetics, and Activity of the Combination of Cevostamab and Elranatamab in Participants With Relapsed or Refractory Multiple Myeloma	IB	RRMM	Cevostamab Elranatamab Tocilizumab	NCT05927571
PF-06863135 As Single Agent And In Combination With Immunomodulatory Agents In Relapse/Refractory Multiple Myeloma	I	RRMM	PF-06863135 monotherapy IV or SC PF-06863135 + dexamethasone PF-06863135 + lenalidomide PF-06863135 + pomalidomide	NCT03269136

**Table 3 pharmaceuticals-16-01628-t003:** Talquetamab-based combination trials.

Trial	Phase	Population	Intervention	Trial ID
MonumenTAL-3	III	RRMM with ≥1 prior line	Talquetamab sc with daratumumab and pomalidomide (Tal-DP) vs. talquetamab, daratumumab (Tal-D) vs. daratumumab, pomalidomide, dexamethasone (DPd)	NCT05455320
TRIMM-3	I	RRMM not candidate for available therapy	Talquetamab and teclistamab each in combination with a Programmed Cell Death receptor-1 (PD-1) inhibitor	NCT05338775
MajesTEC-7	III	NDMM	Teclistamab, daratumumab, lenalidomide (Tec-DR) and talquetamab, daratumumab, lenalidomide (Tal-DR) vs. daratumumab, lenalidomide and dexamethasone (DRd)	NCT05552222
MonumenTAL-2	I	NDMM	Treatment A: talquetamab + carfilzomib Treatment B: talquetamab + daratumumab + carfilzomib Treatment C: talquetamab + lenalidomide Treatment D: talquetamab + daratumumab + lenalidomide Treatment E: talquetamab + pomalidomide	NCT05050097

**Table 4 pharmaceuticals-16-01628-t004:** Cevostamab-based combination trials.

Trial	Phase	Population	Intervention	Trial ID
CAMMA 1	Ib	RRMM	Weekly cevostamab or cevostamab, pomalidonide, dexamethasone or cevostamab, daratumumab, dexamethasone	NCT04910568
CAMMA 2	I/II	RRMM, triple-class refractory, and previously treated with anti-BCMA therapies	Cevostamab	NCT05050097
GO43979	I	RRMM	Cevostamab plus elaranatamab	NCT05927571
Cevostamab following CAR T cell therapy for RRMM	II	RRMM	Cevostamab consolidation following BCMA CAR T cell therapy	NCT05801939

**Table 5 pharmaceuticals-16-01628-t005:** Main adverse events with bispecific antibodies (any grade/grade 3–4).

	MajesTEC-1	MagnetisMM-3	MonumenTAL-1	GO39775
BsAb	Teclistamab	Elranatamab	Talquetamab	Cevostamab
CRS %	72.1(0.6)	57.7 (0)	QW: 79 (2.1) Q2W: 74.5 (0.7) Prior TCR: 76.5 (2)	80.7 (1.2)
ICANS %	3 (0)	3.4 (0)	QW: 11 Q2W: 11 Prior TCR: 3	14.3 (0.6)
Neutropenia %	71.5 (65.5)	48.8 (48.8)	QW: 35 (30.8) Q2W: 28.3 (22) Prior TCR 49 (27.5)	38
Thrombocytopenia (%)	42.4 (22.4)	30.9 (23.6)	QW: 27.3 (20.3) Q2W: 29.7 (18.6) Prior TCR: 37.3 (29.4)	24
Infections (%)	80 (55.2)	69.9 (39.8; grade 5: 6.5)	QW: 58.7 (19.6) Q2W: 66.2 (14.5) Prior TCR: 72.5 (27.5)	45 (20)

## Data Availability

Data sharing is not applicable.

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
