# Peer review of "Novel Immunotherapies and Combinations: The Future Landscape of Multiple Myeloma Treatment"

_pharmaceuticals, 2023, doi:10.3390/ph16111628_

Round 1

Reviewer 1 Report

Comments and Suggestions for Authors

The authors report novel immunotherapies and combinations: the future landscape of multiple myeloma treatment.

1.     The authors should provide a summary graphical abstract such as immunotherapy, myeloma signaling, and treatment.

Author Response

We provided a graphical abstract as suggested

Reviewer 2 Report

Comments and Suggestions for Authors

The manuscript titled "Novel immunotherapies and combinations: the future Landscape of multiple myeloma treatment" discusses the use of antibodies, and their combination with drugs, to treat melanoma cancer. This is an exciting topic, and the manuscript is well-written and organized. However, there are a few concerns that the authors must address in order to improve the manuscript. 

Firstly, several review articles on this topic have been published in recent years, so the authors need to cite them and explain how their work is different from the previous studies. Some examples of these review articles are: "Cancers 2022, 14(3), 778" https://doi.org/10.3390/cancers14030778, "Current Treatment Options in Oncology volume 23, pages1428–1442 (2022)", and "Melanoma Res. 2021 Feb 1;31(1):1-17. doi:10.1097/CMR.0000000000000702".

Secondly, the conclusion and future direction of the manuscript need to be elaborated upon. The authors should discuss the future direction of this research and provide more details on the implications of their findings.

Author Response

  1. Our review regards new generation antibodies as conjugated monoclonal antibodies and bispecific antibodies in the treatment of multiple myeloma and not melanoma
  2. We provided to discuss future direction in the conclusion

Reviewer 3 Report

Comments and Suggestions for Authors

More and colleagues present a thoughtful review of next-generation antibody therapies for relapsed and refractory multiple myeloma.  The work presented is thorough, covering the key trials leading to approval of BiTEs as well as ongoing combination studies with these agents and is of interest to the field. 

1. pg. 2. "these drugs have recently demonstrated superiority of quadruplets over triplets". The MASTER trial showed that 80% of patients achieved MRD negativity and 71% reached two consecutive MRD-negative assessments during therapy. Please present results that compare the trial design, patient selection, measurement of patient outcomes, PFS, OS, TTP.

2. pg. 2. Please include and discuss which trials included MM patients wit HR disease, HR cytogenetics, and clinically HR patients. Trial enrollment and inclusion criteria need to be fairly equitable if you are comparing the outcomes from two trials.

3. Please describe the patient selection and enrollment criteria for MagnaetisMM-1 and 3. Were the patient selection and enrollment criteria comparable. 

-Some paragraphs are very long eg section 5.1's first paragraph is an entire page. Consider revision by a native English speaker.

4. Table: would cut the "Drug" prefix before each line in the intervention tab.

5. Please discuss this Commentary with respect to the reproted real world results. Chakraborty R, Al Hadidi S. Intent Matters: Real-World Applicability of Idecabtagene Vicleucel Usage in the United States. J Clin Oncol. 2023 Jul 10;41(20):3657-3658. 

6. Discuss  treatment with sequential B-cell maturation antigen-targeted therapy.

7. Please discuss how the trials demonstrate such vastly different outcomes. For example,  Sanoyan, D.A., Seipel, K., Bacher, U. et al. Real-life experiences with CAR T-cell therapy with idecabtagene vicleucel (ide-cel) for triple-class exposed relapsed/refractory multiple myeloma patients. BMC Cancer 23, 345 (2023). compared to Munshi NC, Anderson LD, Shah N, Madduri D, Berdeja J, Lonial S, et al. Idecabtagene Vicleucel in Relapsed and Refractory Multiple Myeloma. N Engl J Med. 2021;384(8):705–16.

8. Please discuss responses to BCMA-targeted CAR T-cells after prior BCMA-directed therapy.

Comments on the Quality of English Language

Please see above.

Author Response

  1. We included recently published data of MASTER trial
  2. Really, we have inserted some general concepts without comparison between trials so we think discussion requested by review is beyond the main topic of this review
  3. We described the patient selection and enrollment criteria of MagnetismMM-1 and 3 and we shortened paragraph 5.1
  4. We modified as suggested
  5. We inserted paragraph number 11 to deal with this topic
  6. IN the paragraph 11 we discussed sequential anti-BCMA therapy
  7. According to us Sanoyan, D.A., Seipel, K., Bacher, U. et al. Real-life experiences with CAR T-cell therapy with idecabtagene vicleucel (ide-cel) for triple-class exposed relapsed/refractory multiple myeloma patients. BMC Cancer 23,345 (2023) compared to Munshi NC, Anderson LD, Shah N, Madduri D, Berdeja, Lonial S, et al. Idecabtagene Vicleucel in Relapsed and Refractory Multiple Myeloma. N Engl J Med.2021;384(8):705–16 demonstrated similar results in a real life population and a clinical trial population, whose characteristics are similar, confirming CART applicability also in real life settings where most patients don’t comply with trials eligibility criteria. In this review we didn’t focus on CAR T cells so we didn’t explain different results from CART studies, but we rapidly cited it in the paragraph number 11.
  8. We inserted the new paragraph number 11 to discuss sequential anti-BCMA therapy

Round 2

Reviewer 2 Report

Comments and Suggestions for Authors

The manuscript can accepted in this current form.